# OmniPred: Language Models as Universal Regressors

**Xingyou Song**[1*], **Oscar Li**[2*†], **Chansoo Lee**[1], **Bangding (Jeffrey) Yang**[3], **Daiyi Peng**[1], **Sagi Perel**[1], **Yutian Chen**[1]

[1]**Google DeepMind,** [2]**Carnegie Mellon University,** [3]**Google**
[*]Equal Contribution. [†]Work performed as a student researcher at Google DeepMind.

**Reviewed on OpenReview:** `https://openreview.net/forum?id=t9c3pfrR1X`

## Abstract

Regression is a powerful tool to accurately predict the outcome metric of a system given a set of parameters, but has traditionally been restricted to methods which are only applicable to a specific task. In this paper, we propose OMNIPRED, a framework for training language models as universal end-to-end regressors over $(x, y)$ data from arbitrary formats. Using data sourced from Google Vizier, one of the largest proprietary blackbox optimization databases in the world, our extensive experiments demonstrate that language models are capable of very precise numerical regression using only textual representations of mathematical parameters and values, and if given the opportunity to train at scale over multiple tasks, can significantly outperform traditional regression models.

## 1 Introduction

Regression is a fundamental task for experimental design, in many domains such as hyperparameter tuning, computer software, industrial engineering, and chemical discovery. The goal of regression is to predict a metric $y$ of a general system given a set of input features $x$. Such regressors can later be used for various applications, such as offline optimization (Kumar et al., 2022; Trabucco et al., 2022), online optimization (Cai et al., 2020), low-cost benchmarking (Zela et al., 2022; Eggensperger et al., 2015) and simulation (Mendis et al., 2019; Hashemi et al., 2018; Kaufman et al., 2021).

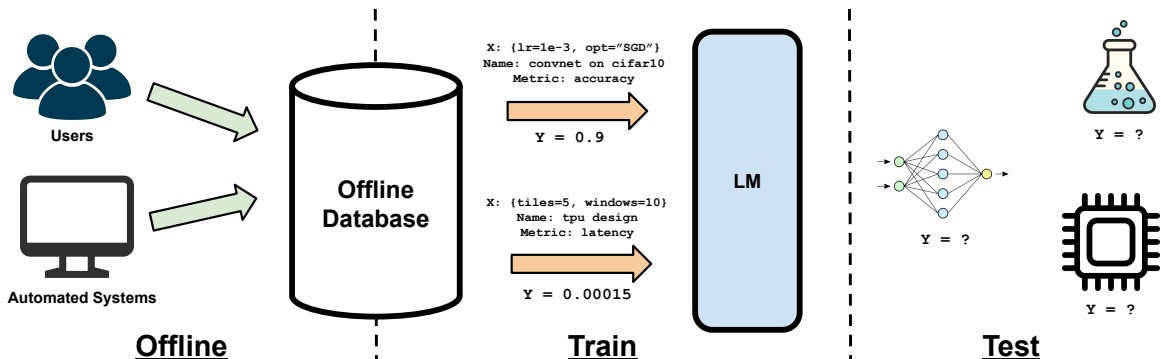

Figure 1: Overview of our method. Using heterogenous offline $(x, y)$ evaluation data collected from a variety of sources, we train a LM-based regressor.

In recent years, large language models (LLMs) have emerged as powerful tools for processing textual representations at scale over massive heterogeneous datasets to represent complex relationships between input features and output labels. Given that LLMs have been shown to be effective for a variety of tasks beyond natural language processing, such as coding (Li et al., 2022), symbolic mathematics (Lewkowycz et al., 2022), and scientific reasoning (Singhal et al., 2022), it is reasonable to wonder: *Can language models be used for numeric regression?*

Answering this question is highly important not only for the traditional field of experimental design, but also for the ever-changing field of LLM research, especially due to recent interest in the ability to forecast outcomes of complex systems (Gruver et al., 2023) and reward modeling in reinforcement learning fine-tuning (Ziegler et al., 2019). The textual processing abilities of LLMs are particularly attractive, as they can potentially bypass the need to tediously featurize inputs (i.e. the $x$'s) into raw numerical tensors. Prior to our work, there has been no such research specifically addressing the feasibility and utility of training a "universal" metric predictor over a large heterogenous offline dataset.

Our core contributions in summary, are as follows:

- We propose OMNIPRED, the first scalable yet simple metric prediction framework based on free-form textual representations, applicable to general input spaces.

- Through only these text and token-based representations, OMNIPRED is capable of very accurate metric predictions over experimental design data.

- By leveraging multi-task learning across vastly different input spaces and objectives, in many cases OMNIPRED can outperform traditional regression models such as MLPs and boosted trees.

- These transfer learning benefits persist even on unseen tasks after online finetuning OMNIPRED on small amounts of new evaluation data.

## 2   Related Work and Motivation

Traditional regression methods have widely used statistical techniques such as Gaussian Processes (GPs), tree-based methods, and multilayer perceptrons (MLPs), to predict a scalar objective given a fixed-length feature vector, commonly seen in tabular data settings. Multitask (Bonilla et al., 2007) and contextual (Krause & Ong, 2011) variants have been further proposed for transfer learning purposes, but still require fixed-length tensor representations of $x$, and can thus only use previous $x$ from the same input space. Additional recent works utilizing deep learning-based regressors include Transformers (Hollmann et al., 2023; Huang et al., 2020; Garg et al., 2022), recurrent neural networks (Hashemi et al., 2018), graph neural networks (Lukasik et al., 2020; Gao et al., 2023), and deep-hierarchical GPs (Fan et al., 2024), which allow length-independence. Even so, a frequent issue is still the reliance on *tensor representations* of $(x, y)$.

Tensor representations are commonly *problem-dependent*, where each tensor element may need to be in a reasonable numerical range (e.g. in $[-1, 1]$) as inputs to a model. Thus to represent $x$, every categorical feature must be one-hot embedded against user-provided choices, and scalar features may need to be rescaled against user-provided bounds. Dynamic yet minor input space changes such as new bounds or additional categories, are incompatible with this static representation. To represent $y$, a raw objective in $\mathbb{R}$ may also need to be rescaled, which can be problematic at test-time when encountering outlier $y$-values. Dealing with this issue leads to implementing complicated nonlinear warpings (Daimon, 2011; Yeo & Johnson, 2000), many of which are also data-dependent (e.g. require storing min/max values from training data).

| Regressor | Dynamic Input Spaces? | Can Multitask? | Tensorize $x$? | Rescale $y$? |
|---|---|---|---|---|
| MLP | No | Only fixed spaces | Yes | Yes |
| Tree-based | No | Only fixed spaces | Yes | Optional |
| Gaussian Process (GP) | No | Only fixed spaces | Yes | Yes |
| GNN / Transformer / RNN | No | Only fixed domains | Yes | Yes |
| OMNIPRED (Ours) | **Yes** | **Yes** | **No** | **No** |

Table 1: Comparisons between the flexibilties of different typical regressors.

These issues are summarized in Table 1. In principle, an ideal regressor should process $x$ and output $y$, both **in absolute terms**, independent of changing external statistics or search constraints. For example, if the underlying relationship is $y = x^2$, then the regressor's prediction at $x = 2$ should be the same regardless if the constraint is $x \in [1, 5]$ or $x \in [0, 100]$. One way to accomplish this is to represent $x$ with an independent universal tokenizer without problem-specific numeric transformations (Zhou et al., 2023). This immediately

unlocks a large amount of transferrability when dealing with variable-length inputs and additional contextual metadata.

Previous works in the token-based, or effectively text-to-text paradigm, have mostly been in reinforcement learning from human feedback (Ziegler et al., 2019), where regression over textual responses (the "$x$"), also known as *reward modelling*, has been crucial to the success of recent interactive LLMs such as ChatGPT (OpenAI, 2022) and Gemini (Google, 2024). While such works have shown that LLMs are able to imitate human ratings in the form of ordinal variables and their softmax probabilities (Bradley & Terry, 1952), they have not shown whether LLMs are capable of regression over precise numeric-based data where $y \in \mathbb{R}$.

This is because the overwhelming current focus has been on subjective *human-based* feedback needed for determining aspects such as creativity, safety, and personality, which contain high aleatoric uncertainty and do not require high-precision measurements. Much less attention has been given towards feedback from complex and natural systems common to experimental design. Given multiple works (Hendrycks et al., 2021; Nogueira et al., 2021) demonstrating their brittle and unreliable numeric abilities, it is non-obvious that language models are capable of high-precision numerical prediction over token-based representations. This is a crucial technical challenge which our paper resolves in the quest for a general-purpose predictor.

## 3 Methodology

### 3.1 Preliminaries and Problem Definition

Since regression is commonly used in blackbox optimization settings, we adopt standard terminology (Golovin et al., 2017; Liaw et al., 2018) from the field. For a given task $\mathcal{T} = (\mathcal{X}, f, \mathcal{D}, m)$, we assume there is a mapping $f : \mathcal{X} \to \mathbb{R}$ for which we obtain *trials* $(x, y)$ from evaluating an input $x$, selected from a (possibly implicit) input space $\mathcal{X}$. We define a *study* as an offline collection of trials $\mathcal{D} = \{x_s, y_s\}_{s=1}^T$. To distinguish between different tasks, there may be observable task-level metadata $m$, which can additionally characterize the task and potentially even describes the behavior of the corresponding mapping $f(\cdot)$.

The goal of standard regression is to obtain a distribution mapping $s : \mathcal{X} \to \mathcal{P}(\mathbb{R})$ such that $s(x)$ accurately approximates $f(x)$ over some distribution of inputs $x \in \mathcal{X}$, provided that a training set $\mathcal{D}^{train}$ is given. In our particular case, we also provide our language model with multi-task training data $\cup \{\mathcal{D}_1^{train}, \mathcal{D}_2^{train}, ...\}$ from other tasks $\{\mathcal{T}_1, \mathcal{T}_2, ...\}$. While these extraneous tasks contain different objectives $f_1, f_2, \ldots$ and may even have different input spaces from each other, training on such additional extraneous data may still lead to transferrability, especially for similar tasks.

A common way to measure the accuracy of predictors (deterministic or probabilistic) is to compute the gap between a final pointwise prediction against the true objective value $y$. For probabilistic regressors $s : \mathcal{X} \to \mathcal{P}(\mathbb{R})$, we may aggregate by e.g. taking the median or mean of the distribution. Since different studies can have vastly different objective scales (e.g. CIFAR10 accuracies are within $[0, 1]$ while synthetic objectives are within $[10^2, 10^9]$), we must therefore normalize the difference based on per-study statistics, i.e. for a specific task, we define the study error as a normalized mean absolute error (MAE):

$$\frac{1}{y_{\max} - y_{\min}} \frac{1}{|\mathcal{D}^{test}|} \sum_{(x,y) \in \mathcal{D}^{test}} |\text{Aggregate}(s(x)) - y| \tag{1}$$

To prevent outlier predictions from significantly swaying average errors, we further clip error maximums to 1.0, equivalent to when the regressor simply outputs boundary values from $\{y_{\min}, y_{\max}\}$.

### 3.2 Language Model

We use a standard language model in which the model observes a prompt and decodes a response. For the regression setting, naturally these correspond to $x$ and $y$ respectively. However, in order to allow multi-task training, the task-specific metadata $m$ must be appended to the prompt in order to distinguish between different tasks, and thus for a given task, the prompt is actually $(x, m)$.

For simplicity, we train a relatively small 200M parameter T5 encoder-decoder (Raffel et al., 2020) *from scratch* to avoid any confounding effects from typical generative language pre-training. We wish to learn a

single set of weights $\theta$, which can be used to form a stochastic regressor $s_\theta : \mathcal{X} \to \mathcal{P}(\mathbb{R})$ given any arbitrary task $\mathcal{T}$. In contrast to settings such as (1) traditional regression requiring training a separate model $\theta_i$ for each task $\mathcal{T}_i$ or (2) requiring completely evaluated trajectories over specialized $x$-tokenizations for in-context learning (Chen et al., 2022; Hollmann et al., 2023), our setting maximizes the usage of training data, much of which may contain unfinished trajectories or free-form $x$-formats.

**Representation:** As mentioned, since input spaces $\mathcal{X}$ and $y$-scales may vary wildly across different tasks, multi-task training requires $x$ and $y$ to be represented in absolute formats independent of the input space and numeric scaling of the specific study. Thus, we express $x$ in a *key-value* format, directly mapping parameter names to values, but do *not* represent[1] the input space $\mathcal{X}$, allowing generalizability to conditional parameters and dynamic constraints. We represent $y$ with custom tokens to guarantee proper decoding via constrained decoding, using specific tokens to express sign, exponent, and significant digits, although more sophisticated alternatives (Koo et al., 2024) are available. An example is illustrated in Table 2. Ablations over different tokenizations are conducted in Appendix A.1.

| | Language Model Textual Representation |
|---|---|
| $x$ | `batch_size:128,kernel:'rbf',learning_rate:0.5,model:'svm',optimizer:'sgd'` |
| $m$ | `title:'classification',user:'some-person',description:'spam detection',`
`objective:'accuracy (%)'` |
| $y$ | `<+><7><2><5><E-1>` |

Table 2: Textual representations used for OMNIPRED. `<*>` represents a single custom token. Input space and $x$ is the same as in Figure 2. Example $y$-tokenization represents a value of $725 \times 10^{-1} = 72.5$.

**Training:** We apply standard Prefix-LM training (Raffel et al., 2020), in which for a given (prompt, response) pair, cross-entropy losses are only computed over response tokens. Pairs are sampled from training data over multiple tasks. One could additionally make the loss more metric-aware by weighting specific tokens or even reinforce with non-differentiable scores, although we maintain simplicity in this paper for now by using uniform cross-entropy. Thus the model will implicitly learn numeric distances from training data.

**Sampling and Decoding:** Through regular temperature decoding, we can repeatedly sample $\widehat{y} \sim s_\theta(x)$, to approximate the prediction distribution over $\mathbb{R}$. To remain robust to strong outliers, instead of using empirical mean, our Aggregate$(\cdot)$ function is defined as the empirical median, since we found it leads to lower error from ablations in Section 6.1. Since the model may need to predict over unseen regions of the input space, we can also assess the model's uncertainty by observing the concentration of sampled predictions $\widehat{y}$ and additionally specific log probabilities across every decoded token.

**Online Finetuning:** To adapt to an unseen task $\mathcal{T}_u$ common for Bayesian optimization settings (Wistuba & Grabocka, 2021), the model can further be quickly finetuned online over the tasks's corresponding training data $\mathcal{D}_u^{train}$, optionally using LoRA (Hu et al., 2022). Finetuning may also help to refocus over seen data, when the model is not fully optimized against a specific study, e.g. if the pretraining dataset was too large.

Appendix B contains additional implementation details such as the specific architecture and vocabulary used.

## 4 Data

Many industries possess large collections of metric data from experiments or systems. However, such data is typically not open-sourced as official training datasets for research. A natural dataset to use may come from multiple hyperparameter optimization trajectories, which tend to have $(x, y)$ evaluations from expensive experiments, expressed as blackbox objectives $y = f(x)$.

### 4.1 OSS Vizier Format

The abstractions in Section 3.1 above are concretely implemented in Open Source (OSS) Vizier (Song et al., 2022), a research interface for blackbox and hyperparameter optimization. Every space $\mathcal{X}$ is a Cartesian

---

[1]In applications such as code search, it is even infeasible to express the space of all possible programs.

product of *parameters*, each of type `DOUBLE` (bounded continuous interval), `INTEGER` (bounded integer set), `DISCRETE` (ordered set of real numbers), or `CATEGORICAL` (unordered set of strings). Every parameter may also have *child parameters*, only active when the corresponding parent parameter is a specific value (e.g. in Figure 2, "beta" is active only if a parent categorical parameter selects "Adam", but not "SGD"). An $x \in \mathcal{X}$ can be expressed as a tabular feature, allowing baseline comparisons against traditional regression models.

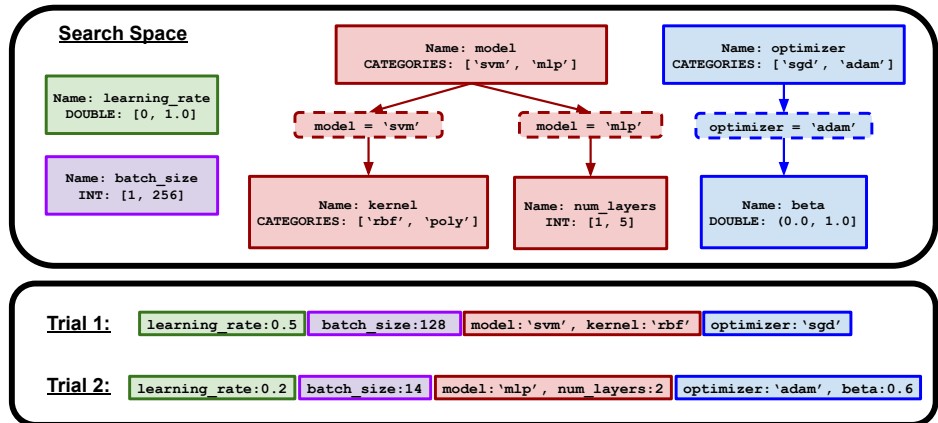

Figure 2: Common example of a (possibly nested) space and suggestions $x$ in OSS Vizier.

Task-level metadata $m$ consists of a title, username, description, objective name, and optional free-form text. Since the OSS Vizier API is meant to provide an optimization service for users, there can be many sources of transferrability due to user-specific settings. These include:

- A single user or team regularly tuning similar experiments.
- Multiple different users tuning similar experiments (e.g. training ResNets on CIFAR10).
- Similar parameters used across different experiments (e.g. "learning rate").
- Metadata $m$ describing the nature of the objective function.

### 4.2 Datasets

**BBOB (Shifted):** For precise controlled experiments where we can generate synthetic datasets and perform online evaluations, we create a multi-task version of the BBOB benchmark (ElHara et al., 2019) containing 24 different synthetic functions, by applying random domain shifts $c$ to transform a vanilla synthetic $f(x)$ into $f(x - c)$, and ranging the dimension over $[2, 6]$. Thus each task $\mathcal{T}$ is parameterized by $m = $ (function class, dimension, shift), and the corresponding objective can be seen as $f(x, m)$, allowing evaluation over unseen $m$. For a specific task $\mathcal{T}_i$, we minimize the *in-study training data* size $\mathcal{D}_i^{train}$ but freely vary *inter-study* training data $\{\mathcal{D}_j^{train}\}_{j \neq i}$ from different tasks $\{\mathcal{T}_j\}_{\neq i}$. Thus traditional regressors (e.g. MLPs) which can only train from a single $\mathcal{D}_i^{train}$ will struggle to regress the corresponding $f_i$ under this limited data condition. In contrast, the LM may perform better as it will have seen trials from other tasks whose functions share similarities with $f_i$.

**Real World Data:** To investigate metric prediction over a rich variety of tasks, we will use data collected by Google Vizier (Golovin et al., 2017). Because we are not limited to training on fully completed trajectories over flat input spaces, we can train on more data than just the 750K studies used in the OptFormer (Chen et al., 2022), as seen from Table 3.

| Property | Statistic |
|---|---|
| # Studies | $\mathcal{O}(70M+)$ |
| # Trials | $\mathcal{O}(120B+)$ |
| # Distinct Users | $\mathcal{O}(14K)$ |

Table 3: Relevant statistics on the real world database. We provide order estimates as there may be numerous ways to define e.g. "legitimate" studies or trials. See Appendix D for further details and data breakdown.

Since the offline dataset is collected from users' blackbox optimization trajectories, we for the most part do not have online access to an actual objec-

tive $f(x)$, rather only data samples $\mathcal{D}$, and thus we must evaluate our predictor's accuracy via a test set $\mathcal{D}^{test} \subset \mathcal{D}$. Thus, we need to take into account how much $\mathcal{D}_{train}$ sufficiently covers the space $\mathcal{X}$, which affects the difficulty of achieving high accuracy on the task. Influencing factors include:

- Trial count: Users can decide when to stop tuning, and thus the size of a study can be on the order of $10^0$ to $10^5$.
- Diversity of trials: By default, a study's trials $\{(x_1, y_1), ..., (x_T, y_T)\}$ form the trajectory of an optimization loop, and thus later trials may converge towards a single local optimum.
- Space size: Approximate cardinality of a space $\mathcal{X}$ is exponential with respect to parameter count, and thus large input spaces will naturally be less explored.

While we apply practical processing steps such as (1) setting a maximum initial trial limit per study and (2) randomly shuffling the trials and then (3) deciding on a fixed train/validation/test splitting ratio (default $0.8/0.1/0.1$), we cannot fully control whether each $\mathcal{D}$ saturates its space $\mathcal{X}$, or essentially how "easy" the task is. Instead, we use a baseline regressor trained only on $\mathcal{D}^{train}$ and evaluated on corresponding $\mathcal{D}^{test}$ as a proxy metric of the difficulty of the task.

## 5 Experiments

We answer the following key questions:

1. Can we simultaneously regress on multiple tasks of different input spaces and objective scales?
2. Are there benefits to multi-task training and are textual signals useful for transfer learning?
3. Does online finetuning improve accuracy over unseen studies outside of the pretraining set?

### 5.1 Simultaneous Regression

In Figure 3, we visually present how a BBOB-trained model captures the overall shape of analytical functions of vastly different objective scales with high precision. Furthermore, the model is capable of expressing uncertainty estimates via i.i.d. prediction samples.

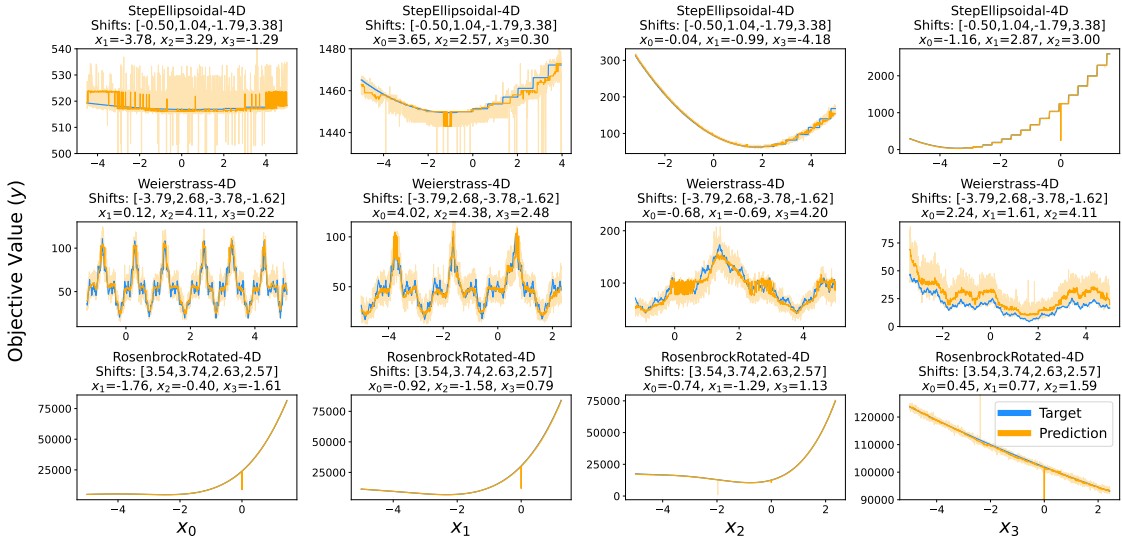

Figure 3: Model prediction samples over selected 4D BBOB functions with unseen shifts. Empirical mode (bolded) and min/max are shown from 10 samples. Over all BBOB functions, we vary the coordinate value $x_i$ while keeping others $x_{j \neq i}$ fixed.

In Figure 4 for a model trained over real world data, we present an analogous visualization over hand-selected studies with drastically different input spaces, representative of objectives tuned commonly in real world settings. These include standard machine learning (e.g. image classification and language modeling),

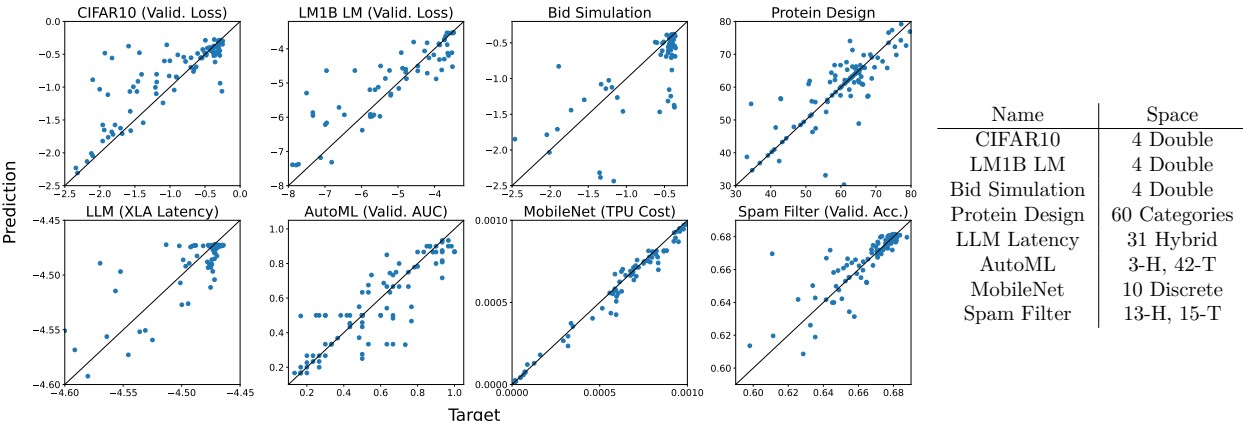

| Name | Space |
|------|-------|
| CIFAR10 | 4 Double |
| LM1B LM | 4 Double |
| Bid Simulation | 4 Double |
| Protein Design | 60 Categories |
| LLM Latency | 31 Hybrid |
| AutoML | 3-H, 42-T |
| MobileNet | 10 Discrete |
| Spam Filter | 13-H, 15-T |

Figure 4: **Left:** Diagonal fit (/) is better. Model's $y$-prediction vs. ground truth over varying studies. Corporate-specific objective names are redacted. **Right:** Corresponding input spaces. "#-H, \$-T" is shorthand for a conditional hybrid input space with # root parameters and \$ total possible parameters.

production systems (e.g. bid simulation, LLM inference latency), and scientific research (e.g. protein and hardware design).

## 5.2 Multi-task Transferrability

In this subsection, we demonstrate the model's ability to transfer learn, i.e. improve accuracy over a specific task using knowledge gained from other similar but non-equivalent tasks, in contrast to "single-task" regressors (described in Appendix C) which only observe training data from the task being evaluated. Note that single-task baselines such as MLPs are incapable of multi-task training when tasks have different dimensionalities.

In Figure 5, we clearly see that the model's accuracy improves with more tasks seen in training and eventually outperforms all traditional baselines. For AutoML studies, the error is averaged from a fixed subset of encountered studies. For BBOB, we can further demonstrate the model's inter-study generalization capabilities over metadata $m$ (as opposed to $x$) by evaluating on unseen tasks with new shifts not encountered during training.

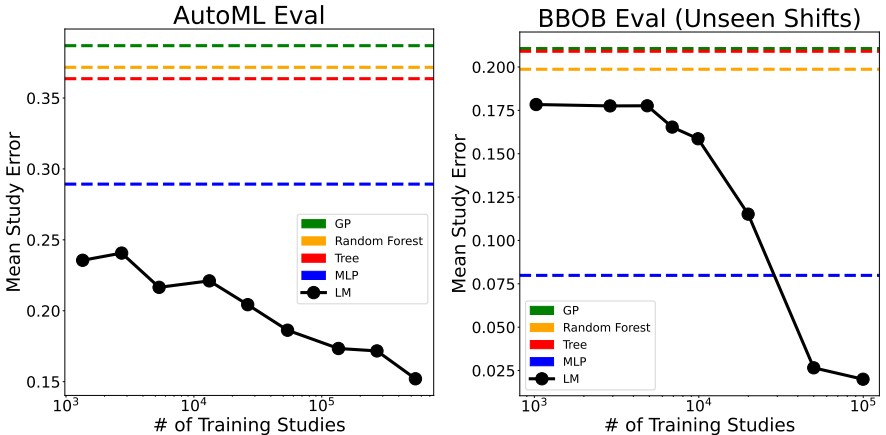

Figure 5: Lower (↓) is better. Mean study prediction error of the model when varying the amount of different studies used in training (log scale). Colored horizontal lines display single-task baseline errors.

To verify whether the model is performing transfer learning by reading textual cues, in Table 4 we compare results against the case when data is "anonymized" using a study-dependent hash function. For BBOB, we hash metadata $m$ which originally displayed (function class, dimension, shift). For AutoML, we hash parameter names and string values. Each study can still be uniquely identified and trained on, but the model can no longer observe useful correlations from common textual clues. Interestingly, the model fails to train over the full anonymized BBOB dataset, a case when the data is too large and heterogeneous.

| Datasets (# Training Studies) | Mean Study Error (↓) | |
| --- | --- | --- |
| | Original | Anonymized |
| BBOB (50K) | **0.03** | 0.46 |
| BBOB (Full 1M) | **0.01** | FAIL |
| AutoML (26.3K) | **0.19** | 0.44 |
| AutoML (Full 540K) | **0.15** | 0.43 |

Table 4: Lower (↓) is better. Comparisons between models trained on original vs anonymized data, across BBOB-Shifted and AutoML test trials. "FAIL" means the model failed to even train.

In Figure 6, we further see that for the model, multi-task training consistently improves over single-task training, and in regimes with relatively lower input space saturation (i.e. low trial to parameter count ratio) from training data, multi-task models outperform traditional baselines over several different domains. Interestingly, a single-task model trained from scratch remains a competitive choice and for certain domains such as AutoML, can even outperform all other single-task baselines. We hypothesize this is due to language-based representations being more appropriate for the conditional structures of these domains (e.g. AutoML).

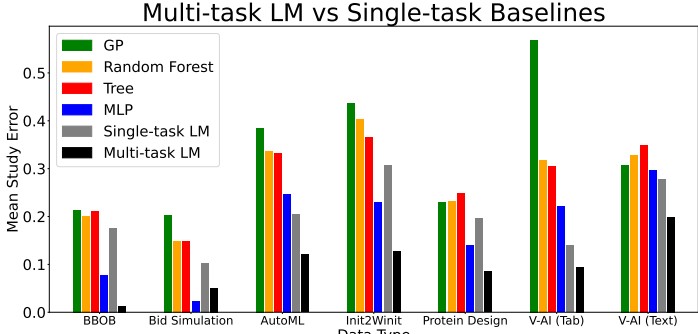

| Name | # Studies | Avg. TpS | Avg. SS |
| --- | --- | --- | --- |
| BBOB | 1M | 30 | 4.0 |
| Bid Simulation | 22K | 698 | 4.6 |
| AutoML | 540K | 250 | (3.3, 29.9) |
| Init2winit | 2K | 176 | 3.6 |
| Protein Design | 54K | 584 | 125.6 |
| Vertex AI (Tabular) | 1.4M | 88 | (4.6, 42.4) |
| Vertex AI (Text) | 544K | 118 | 56.0 |

Figure 6: **Left:** Lower (↓) is better. Aggregate error across different domains. **Right:** Statistics on domains. Shorthand notation: "TpS" = Trials per Study, "SS" = Space Size, with brackets (#, $) denoting conditional space with # root parameters and $ total possible parameters. Note that all baselines are single-task regressors.

## 5.3 Online Finetuning Analysis

We first examine the conditions in which finetuning may be beneficial. In Table 5, we finetune various pretrained models over AutoML studies. While there is negligible benefit in finetuning the AutoML model on its data again, we see that a model pretrained over the entire real world dataset is able to finetune to the same level of accuracy as a pretrained AutoML model, while a BBOB-pretrained model leads to significantly worse results than even a single-task model. This suggests that knowledge obtained from pretraining can have a large (positive or negative) influence on transferrability over specific domains such as AutoML.

| Pretraining Dataset | Mean Study Error (↓) on AutoML | |
| --- | --- | --- |
| | Before Finetuning | After Finetuning |
| None (Single-Task) | 0.98 | 0.20 |
| BBOB | 0.98 | 0.45 |
| AutoML | **0.15** | **0.15** |
| All RealWorldData | 0.31 | **0.15** |

Table 5: Lower (↓) is better. Mean study errors of pretrained models and their corresponding finetuned versions.

We further examine this effect by evaluating over unseen tasks, i.e. those which were newly created after the original training set was scraped, and can contain studies from new users and objectives. In Figure 7,

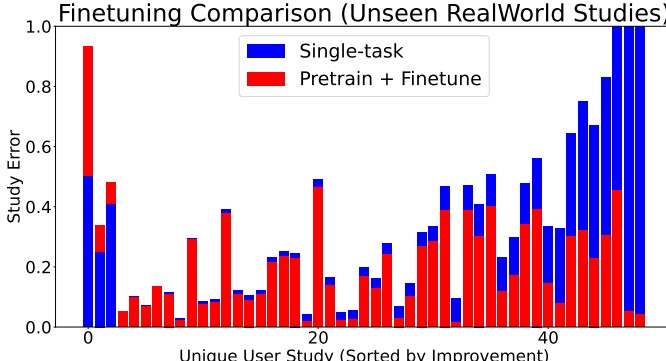

| Method | Mean Study Error (↓) |
|---|---|
| Single-task (LM) | 0.28 |
| Pretrain (LM) | 0.68 |
| Pretrain + Finetune (LM) | **0.21** |
| MLP (Baseline) | 0.25 |
| Tree (Baseline) | 0.32 |
| Random Forest (Baseline) | 0.32 |
| Gaussian Process (Baseline) | 0.42 |

Figure 7: **Left:** Lower (↓) is better. Example LM study errors over unseen studies filtered over random distinct users. **Right:** Aggregate comparisons across different methods over 1000 unseen studies.

we compare initialization from scratch (leading to single-task training) against a pretrained model on older real world data. We see that knowledge obtained from pretraining can significantly transfer over and help predictions over new tasks, although as seen on the left with three studies, there are few cases of negative transfer.

## 6 Ablations

We further ablate certain important settings and scenarios which affect the model's prediction accuracy below. Appendix A contains additional ablations, specifically (1) around our choice of $y$-tokenization and (2) experimental outcomes when using ranking-based regression metrics.

### 6.1 Effect of Sampling

The LM can output extreme outliers in its $y$-prediction, usually due to an inaccurate prediction on the exponent token or significant digits. While such issues do not occur once the model has nearly perfectly regressed on a task (e.g. BBOB), they occur frequently on realistic tasks with nontrivial error (e.g. AutoML) and thus require techniques for correction. One obvious method to reduce error is to increase sample count, as demonstrated in Figure 8.

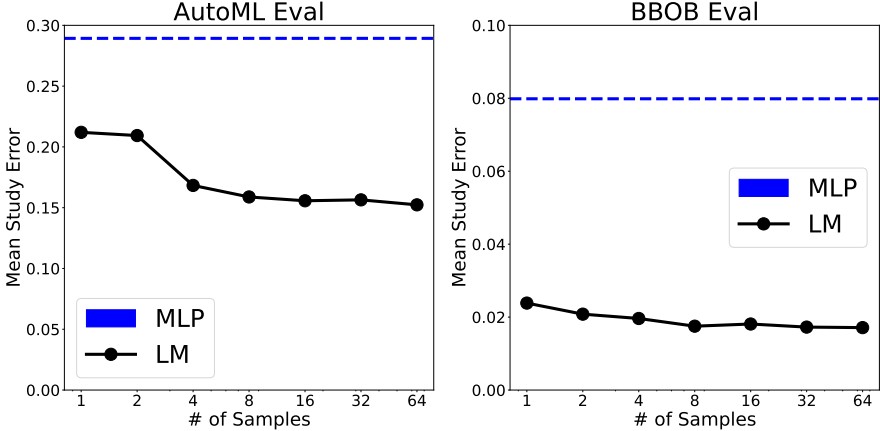

Figure 8: Lower (↓) is better. Mean study error when varying the samples used during inference (log scale).

An additional method is to change the method of aggregation across samples. In Table 6, we see that using the median considerably outperforms both max-likelihood and mean. We hypothesize this is due to the median's robustness to hallucinated outlier samples which can occur with relatively high probability and can also skew the mean.

|  | Mean Study Error (↓) | |
|---|---|---|
| Empirical Aggregation Method | AutoML (Full 540K) | BBOB (Full 1M) |
| Median (default) | **0.15** | 0.01 |
| Max-likelihood | 0.22 | 0.01 |
| Mean | 0.23 | 0.01 |

Table 6: Lower (↓) is better. Comparisons between different aggregation methods when using 64 samples when using full datasets for pretraining.

## 6.2 Uncertainty Calibration

Although the main metric used throughout our work is based on pointwise prediction, an important ability for regressors is to express uncertainty when they are unable to provide an accurate prediction. This is particularly useful in applications such as Bayesian optimization, where uncertainty can be used as an exploration proxy. In this section, we examine whether the model can quantify uncertainty even if we did not calibrate or tune any of the models for such purposes.

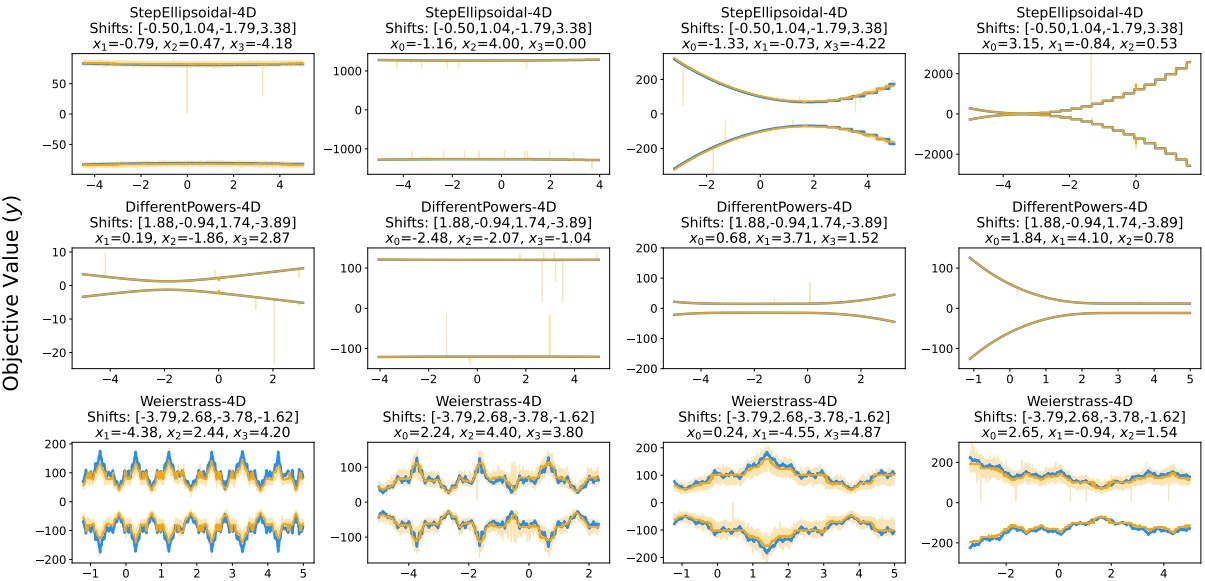

Figure 9: Setting similar to Figure 3, but with bimodality.

We begin by demonstrating the LM's ability to nonparametrically express distributions with multiple modes in Figure 9 when we train the model against randomly sign-flipped BBOB objectives. In contrast, traditional methods such as ensembled MLPs (Havasi et al., 2021) and Gaussian Process mixtures (Bonilla et al., 2007) must specify the mixture count a priori.

|  |  | Pearson, Kendall-Tau, Spearman | |
|---|---|---|---|
| Regressor | Uncertainty Metric | AutoML | BBOB |
| Gaussian Process | Predicted SD | 0.254, 0.230, 0.307 | 0.018, 0.068, 0.048 |
| LM w/ mean aggregation | Sample SD | 0.560, 0.487, 0.625 | 0.360, 0.366, 0.454 |
| LM w/ median aggregation | Harrell-Davis SE | 0.525, 0.412, 0.539 | 0.360, 0.293, 0.380 |

Table 7: Higher (↑) is better. Rank correlation between quantified uncertainty (SD = standard deviation, SE = standard error) and actual error over studies with at least 10 test trials (all BBOB studies and 641 AutoML studies)

Furthermore, we measured the correlation between uncertainty and error on each study. In Table 7, we report the average correlation across studies. Interestingly, although Table 6 demonstrated that mean aggregation over LM samples is worse for prediction than median aggregation, the errors are well correlated with the standard deviation of the samples.

### 6.3 Study Size vs. Multi-task Gain

Intuitively, as a task's space becomes more saturated with trials, single-task training becomes more sufficient for accurate prediction. In Figure 10, we plot the gains from multi-task LM training over the single-task MLP baseline to validate this hypothesis. Gains are maximized at roughly ≈50 training trials and diminish as the number of training trials increases. Note that maximal gains do not occur with ≈0 trials, as presumably *some* training trials are still needed to identify the structure of a task.

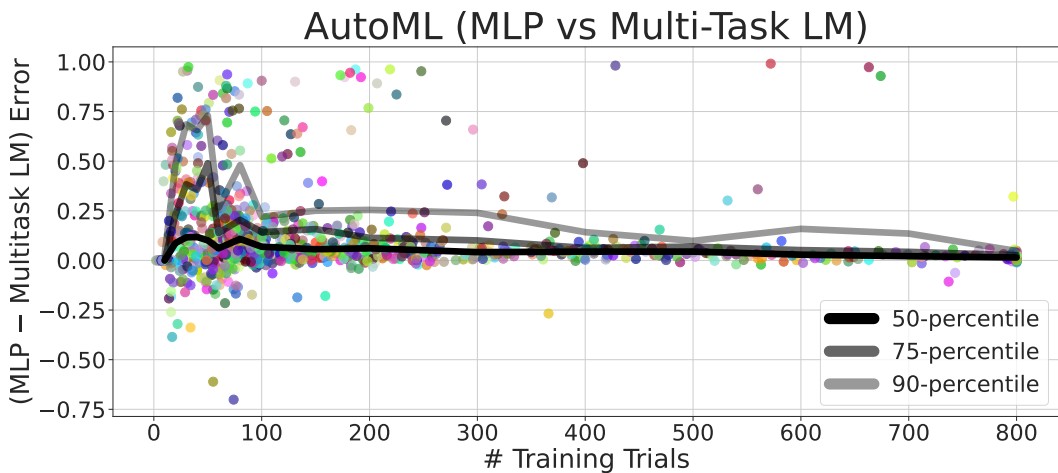

Figure 10: Higher (↑) is better. Study error differences between MLP and multi-task LM over individual AutoML tasks. Percentiles are computed after binning the x-axis appropriately.

## 7 Discussion: Limitations and Extensions

In this work, our emphasis was to demonstrate the promise of applying language modeling to general-purpose regression, and thus our design choices remained relatively simple to avoid confounding factors. We list some limitations of our specific design, which opens many more potential areas of exploration.

**In-Context Learning (ICL):** By design, we maximized the allowed prompt length to allow flexibility in representing $(x, m)$ and thus did not use in-context regression, since a single $(x, m)$ prompt could be 10K+ tokens long for some applications. While online finetuning in principle allows *infinite* amounts of data to be absorbed at inference time, it is worth investigating ICL methods which allow arbitrarily long prompts as well. Current methods (Chen et al., 2022) which require significant input compression are only applicable to tabular-like $x$'s. Additional use of ChatGPT and other service-based chat APIs (Vacareanu et al., 2024) rely on regression as an emergent property of expensive LLM-based training, making their serious use difficult, especially as such services cannot absorb large amount of user-defined offline $(x, y)$ data which often contain the most relevant information for finetuning.

**Hallucinations:** By giving the model the freedom to sample $y$-values over approximately all of $\mathbb{R}$, wildly inaccurate outlier predictions are now possible. This can be exacerbated by a wrong prediction over a significant float token (e.g. leading digit or exponent). Although for convenience, we used an unweighted cross-entropy loss in which all float tokens are of equal importance, prediction accuracy can be improved by weighting more significant tokens, making the training loss more aware of numerical distances over $\mathbb{R}$.

**Prompt-Side Numeric Tokenization:** In this work, we directly represented numeric parameter values from $x$ into the default human readable format (e.g. 1234.5 is serialized simply to '1234.5') to be consistent

with LLM literature. This may be suboptimal, as the corresponding tokens may not exactly be digit-by-digit (e.g. T5's tokenization leads to tokens {'12', '3', '4.5'}). One may instead potentially reuse the custom tokenization for $y$-values (e.g. `<+><1><2><3><4><E0>`) or in text-space, represent using other serializations which emphasize digits atomically, e.g. `'[1 10e2 2 10e1 3 10e0 4 10e-1]'`) as in (Nogueira et al., 2021).

**Pretrained Language Encoder:** Since $x$ includes parameter names and metadata which contain English words, warm-starting from a model pretrained on English text may improve accuracy. However, most checkpoints comparable to our model's size (<1B params) are not pretrained over experimental data and are unlikely to understand the numerical meaning of e.g. `'learning_rate'`. Furthermore, using a pretrained English model introduces numerous confounding technical choices to consider (e.g. whether to freeze the encoder, tune the learning rate, embed additional custom float tokens, and use more English-based representations of $x$ and $m$), but this topic is worth pursuing in the future. In this work, it is already surprising that a relatively small model trained from scratch can still achieve regression, thus suggesting our technique's broad applicability even without English understanding.

**Computational Costs:** Compared to traditional baselines, a language model requires accelerator usage and has a relatively higher computational cost for both training and finetuning, in addition to higher inference times. In this work, we purposely designed the model to minimize costs by using $\approx$200M params which only requires at most 8 GPUs for training and 1 GPU for inference (see Appendix B).

**Other Input Spaces:** The OSS Vizier API primarily focuses on hyperparameter tuning spaces. Traditionally, more complex spaces such as combinatorics and graphs require sophisticated modeling techniques to form regressors, largely in part to difficulties in representing the $x$'s as tensors. In addition, many applications with non-expressible spaces such program synthesis are impossible to traditionally regress over. We believe that text and token-based representations are highly promising and widely applicable to domains previously unexplored in the field of experimental design.

**Other Metadata:** While we performed ablations which anonymized $m$ and parameter names, one can further investigate which types of metadata are particularly useful for prediction. Such metadata could contain *proxy metrics* introduced by previous domain-specific works, such as Jacobian Covariance for neural architecture search (Mellor et al., 2021) and neural-network norms (Jiang et al., 2020) for the study of generalization. The relevant *code* implementing machine learning or programming tasks may be especially important.

## 8   Impact Statement

This research addresses the ability to regress metrics against textual data. Since any textual metadata may be collected, user-specific information may be used, which raises privacy concerns. This would be particularly true on sensitive topics (e.g. predicting metrics related to personal protected characteristics).

Our research does not involve such sensitive topics for regression, as it is performed over blackbox optimization data. We followed ethical guidelines as all users in our proprietary dataset have consented to have their tuning data saved and used for offline analysis. The proprietary real world dataset does not contain any sensitive personal information other than usernames. Furthermore, we do not release any of the trained checkpoints, as it may be possible to reverse-engineer parts of the training data, which can lead to privacy violations and data leakage.

## 9   Conclusion

Our OMNIPRED framework is a first step towards a universal regressor, capable of performing high-precision predictions over objectives of any scale from vastly different input spaces and applications. Its simple and scalable design allows transfer learning from large amounts of offline diverse evaluations, while its single-task variant can still perform competitively against a wide variety of gold-standard baselines. Furthermore, it is capable of adapting to unseen data through finetuning, while still transferring knowledge from previous data. This research lays the groundwork for exciting new potential expansions in the field of experimental design.

## Acknowledgements

We would like to thank Olivier Bachem, Hado van Hasselt, John Jumper, Aviral Kumar, Yingjie Miao, Sebastian Nowozin, Mangpo Phothilimthana, Zi Wang, Scott Yak, and Amir Yazdanbakhsh for useful discussions and Daniel Golovin for continuing support.

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

# APPENDIX

## A    Additional Ablations

We further ablate certain settings and scenarios which affect the model's prediction accuracy below.

### A.1    $y$-Tokenization

There are multiple possible ways to represent a float (e.g. 123.4) using custom tokens. Using examples from (Charton, 2022; Nogueira et al., 2021; d'Ascoli et al., 2022), the following are all possible representations:

- (Default) Separate Sign and Digit-by-Digit: `<+><1><2><3><4><E-2>`

- Merged Mantissa: `<+1234><E-2>`

- Exponent Before Mantissa: `<+><E-2><1><2><3><4>`

In Table 8, we see that these tokenization differences do not matter with large training data (e.g. multi-task), but matter very much in low data regimes (e.g. single-task). The poor accuracy using "Merged Mantissa" is especially apparent as it requires large amounts of data to learn differences between 18K possible mantissa tokens.

| | AutoML | | BBOB | |
|---|---|---|---|---|
| Tokenization Method | Single-Task | Multi-Task | Single-Task | Multi-Task |
| Default | 0.21 | 0.15 | 0.17 | 0.01 |
| Merged Mantissa | 0.73 | 0.15 | 0.41 | 0.01 |
| Exponent Before Mantissa | 0.24 | 0.15 | 0.17 | 0.01 |

Table 8: Mean Study Error (↓) comparisons between different tokenization methods.

### A.2    Ranking and Correlation Metrics

Although our paper focuses on pointwise predictions which are maximally informative, we can trivially bootstrap our predictions into ranking-based metrics, which may be of downstream use for evolutionary algorithms which are agnostic to $y$-scaling. We see that in general, the multi-task LM generally maintains competitive ranking metrics.

| | Kendall-Tau, Spearman Correlation (↑) | | | | | | |
|---|---|---|---|---|---|---|---|
| Regressor | BBOB | Bid Simulation | AutoML | Init2Winit | Protein Design | V-AI (Tab) | V-AI (Text) |
| Gaussian Process | 0.69, 0.80 | 0.80, 0.91 | 0.04, 0.06 | 0.15, **0.81** | 0.35, 0.43 | -0.03, -0.05 | 0.30, 0.39 |
| Random Forest | 0.59, 0.75 | 0.71, 0.84 | 0.45, 0.57 | 0.55, 0.67 | 0.40, 0.52 | 0.56, 0.71 | 0.29, 0.38 |
| Tree | 0.60, 0.74 | **0.82, 0.93** | 0.37, 0.48 | 0.59, 0.71 | 0.44, 0.57 | 0.55, 0.70 | 0.28, 0.36 |
| MLP | 0.63, 0.76 | 0.73, 0.85 | 0.37, 0.49 | 0.53, 0.63 | 0.47, 0.60 | 0.50, 0.64 | 0.25, 0.34 |
| Single-task LM | 0.01, 0.01 | 0.19, 0.28 | 0.21, 0.28 | 0.05, 0.08 | 0.15, 0.20 | 0.18, 0.24 | 0.11, 0.16 |
| Multi-task LM | **0.92, 0.96** | 0.70, 0.84 | **0.61, 0.73** | **0.65, 0.74** | **0.72, 0.81** | **0.57, 0.72** | **0.49, 0.58** |

Table 9: Higher (↑) is better. Ranking metrics across different regressors and tasks.

# B    Model Details

## B.1    Pretraining

We pretrained our model using T5X (Raffel et al., 2020), which can be found in the open-source codebase `https://github.com/google-research/t5x`. Important hyperparameters, most of which are defaulted, include:

- Architecture size: 12 encoder layers, 12 decoder layers, 12 heads, 64 head dimension, 768 embedding dimension, 2048 MLP dimension.

- Optimizer: Adafactor with base learning rate 0.01 and square root decay. Batch size 256.

- Vocabulary and Tokenizer: SentencePiece tokenizer (Kudo & Richardson, 2018) with a vocabulary of 32000 subword tokens, in addition to the custom tokens for representing $y$-objectives.

- Early stopping: We train for a maximum of 1 million steps, but early stop based on validation loss if overfitting is detected.

The model ($\approx$ 200M parameters) was pretrained using a 4x4 TPU V3.

## B.2    Local Training

During local training, data comes from a single study's limited trials (at most 1000). The training set size can be lower than the batch size (256), and thus we must define one epoch as seeing the training data once, i.e. only one gradient step if training size $\leq$ batch size, but multiple gradient steps otherwise.

We use the same settings from pretraining for consistency, but allow a maximum of 30 epochs. For early stopping, validation loss is now measured over the entire validation set instead of sampled batches. Further specific changes:

- **Single-task training:** Since the model is initialized randomly, we use a larger constant learning rate of $10^3$, consistent with early learning rates encountered during pretraining.

- **Finetuning:** We reload the weights in addition to the optimizer state (containing e.g. momentum parameters) from a checkpoint. We use a smaller fixed learning rate of $10^{-5}$, which is 10x lower than the $\mathcal{O}(10^{-4})$ learning rate normally encountered during late stages of training.

Due to the small training set and relatively low finetuning steps, we used a single 1x1 TPU V3.

## B.3    Inference

At inference time, we perform temperature sampling with a temperature of 1.0. We restrict the logits to only decode the custom floating point tokens for representing $y$-values. To maximize batch size for a 1x1 TPU V3, we generate 64 samples and select the empirical median of these floating point samples as our final prediction when computing prediction error.

# C Baseline Details

## C.1 OSS Vizier Input Space

The space is defined as a list of `ParameterConfig`s, each of which is one of the four primitives:

- `DOUBLE:` Specifies the search range $[l, u]$.

- `DISCRETE:` Specifies a finite subset of $\mathbb{R}$.

- `INTEGER:` Specifies an integer range $[l, u]$.

- `CATEGORICAL:` Specifies a set of strings.

Numeric (`DOUBLE, DISCRETE` and `INTEGER`) parameters may specify optional log or reverse-log scaling. The log scaling is most commonly used for tuning learning rates.

## C.2 Data Processing for Flat Space

A *flat space* is where every trial in the study specifies every parameter configured in the space. In this case, we convert the parameters into the unit hypercube $[0, 1]^d$. For numeric parameters, we scale all values into the range $[0, 1]$, by default using linear scaling unless (reverse)-log scaling was configured. For `CATEGORICAL` parameters, we use a one-hot encoding.

## C.3 Data Processing for Conditional Space

A *conditional space* occurs when one parameter may be unused, depending on its parent parameter's value. Conditional spaces commonly appear in AutoML settings where different model classes require a different set of parameters to be tuned. Another common use case is when we wish to optimize a numeric hyperparameter in the log scale but include 0 in the search (e.g. dropout rate, regularization coefficient), i.e. $\{\texttt{UNUSED}\} \cup [l, u]$ where $l > 0$.

For a categorical parameter, we simply add an extra out-of-vocabulary dimension for the one hot encoding.

For a numeric parameter, we first convert parameter values to $\texttt{NaN} \cup [0, 1]$, using the same scaling as in the flat space but mapping all `UNUSED` to `NaN`. We then add a custom layer (one per parameter) which is defined as:

$$x \mapsto \begin{cases} v_p & \text{if } x \text{ is } \texttt{NaN}, \\ x & \text{otherwise} \end{cases}$$

where $v_p$ is a parameter that is trained together with the rest of the model.

## C.4 Regressor Baselines

Below, we list out the specific implementation details of our regressor baselines. One nuanced issue is of hyperparameter tuning the regressors themselves, which could affect results. In order to be reasonably fair to all regressors (including our own OmniPred which has its own hyperparameters), for each regressor, we used a reasonable fixed set of hyperparameters for consistency throughout all experiments.

We emphasize that our paper's contributions are mostly on regression using flexible string representations and large-scale multi-task training, and do not claim to replace widely accepted baselines in single-task, apples-to-apples comparisons.

**Gaussian Process:** The GP regressor model is from the GP-Bandit implementation (Song et al., 2024) found in Open Source Vizier at `https://github.com/google/vizier` and consists of the following:

- $\alpha \sim$ TruncatedLogNormal controls the amplitude of Matern-5/2 kernel.

- $\lambda_i \sim$ TruncatedLogNormal (i.i.d. for each dimension $i$) controls the length scale for the $i$-th dimension.

- $\sigma \sim$ TruncatedLogNormal controls the Gaussian noise.

- $z \sim$ Normal$(0, \sigma)$ is the observation noise.

- $f \sim$ GP$(\lambda, \alpha)$ is the function.

- $y \sim f(x) + z$ is the noisy function.

The algorithm then uses L-BFGS to obtain the MAP estimate of $\alpha, \lambda$ and $\sigma$.

One caveat here is that this model requires a non-linear preprocessing on the observations and thus predicts $y$ in the preprocessed space. This preprocessing is found to be critical to achieving stable regression across Vizier studies, which have a wide variety of value ranges. Since the preprocessing is non-linear, we cannot obtain the predictive distribution over the raw observation in a closed form. Instead, we take 1000 samples from the GP, apply the inverse of the preprocessor, and then take the mean.

**Tree and Random Forest:** We use the standard API (`XGBRegressor`, `XGBRFRegressor` in `https://github.com/dmlc/xgboost`) found in XGBoost (Chen & Guestrin, 2016), with the following grid-search hyperparameter sweeps for each study:

- `"min_child_weight"`: $[1, 5, 10]$
- `"learning_rate"`: $[0.001, 0.01, 0.1]$
- `"gamma"`: $[0.0, 0.3, 0.5]$
- `"subsample"`: $[0.6, 0.8, 1.0]$
- `"colsample_bytree"`: $[0.6, 0.8, 1.0]$
- `"max_depth"`: $[3, 5, 7]$

Although tree-based methods do not generally require rescaling, we still applied consistent $x$-preprocessing (in particular to deal with optional log or reverse-log scaling).

**Multilayer Perceptron:** The base architecture consists of a 2-layer ReLU dense network of hidden size 256 with a final scalar output. $y$-values are normalized using `tf.keras.layers.Normalization` which subtracts mean and divides by standard deviation computed empirically from the training data. Training was performed with an Adam optimizer using learning rate $10^{-2}$, full batch training over 100 epochs, and mean squared error.

# D   Google Vizier Data

## D.1   Study Preprocessing

Since Google Vizier is a service in which users control evaluations, much of the raw study data can be quite chaotic. We apply certain preprocessing techniques to make the data more conducive to training and evaluation.

**Removing bad trials:** Users may ignore or fail to evaluate a proposed $x$. Furthermore, during some trials, the $y$-objective may be denoted with a special "infeasible" value (e.g. if a high batch size led to GPU out-of-memory errors, or if training encountered NaNs). We remove such trials from consideration in our work, although one can extend our $y$-tokenization to support infeasibility in later works.

**Trial count hard limit:** Some raw studies can contain trials in upwards of $10^5$ trials, which could dominate the data distribution. We therefore apply a hard limit and only consider the first $10^3$ trials per study.

**Filtering specific users:** There are specific human and automated "power users" which produce orders of magnitude more studies than the average user. Some automated users in particular are simply automatic unit tests involving the use of OSS Vizier. We disregard studies from these users to prevent them from dominating the data distribution.

## D.2   Real World Data Descriptions

**(Overall) Entire Database:** No filters were applied. All studies from the database were exported on March 31, 2023. Finetuning experiments involving unseen studies consist of studies created after this date.

**Bid Simulation:** Contains hyperparameters for proprietary bid simulation. The simulator estimates how advertisements might have performed in terms of key metrics such as cost, impressions, clicks, and conversion volume.

**AutoML:** Collection of proprietary AutoML data for tuning user objectives.

**Init2Winit:** Data from running deterministic, scalable, and well-documented deep learning experiments, with a particular emphasis on optimization and tuning experiments (e.g. ResNets on CIFAR10, Transformers on LM1B). Public codebase can be found in `https://github.com/google/init2winit`.

**Protein Design:** Each space consists of 50+ parameters, each of which denotes a categorical protein building block.

**Vertex AI (Tabular and Text):** A Vertex AI platform for automated ML model selection and training for tabular or text data. For tabular data, Vertex AI searches over a tree of model and optimizer types, their hyperparameters, data transformation, and other components in the ML pipeline. For text, Vertex AI trains an ML model to classify text data, extract information, or understand the sentiment of the authors. For more information, see:
`https://cloud.google.com/vertex-ai?#train-models-with-minimal-ml-expertise`.

### D.3 Serialization Examples

For transparency, we provide examples of text representations seen by the model. **Disclaimer:** Due to privacy policies, we redacted (in red) some parameter names and values.

| Dataset | Example $x$ | Example $m$ |
|---|---|---|
| AutoML | batch_size: 128
model_type: REDACTED
activation_fn: "tanh"
batch_norm: "True"
dropout: 0.143
embedding_combiner: "mean"
gradient_clip_norm: 1.63e+03
num_hidden_layers: 1
hidden_units[0]: 359
optimizer_type: "AdamOptimizer"
beta1: 0.9
beta2: 0.999
learning_rate: 0.926
nlp_vocabulary_strategy:
    "adjusted_mutual_info"
vocabulary_strategy:
    "adjusted_mutual_info" | title: "n-w597ng99i7lj0-q40zcboi1ea7l"
user: REDACTED
description: ""
objective: "val_categorical_cross_entropy"
amc_model_version: REDACTED
task_type: "multi_class_classification" |
| Init2Winit | dropout_rate: 0.6
decay_factor: 0.0379
label_smoothing: 0.378
lr_hparams.base_lr: 0.00285
lr_hparams.decay_steps_factor: 0.854
lr_hparams.power: 1.94
opt_hparams.one_minus_momentum: 0.0557 | title: "d_sp1-lm1b_trfmr-b1024-2021aug20"
user: REDACTED
description: ""
objective: "valid/ce_loss" |
| Protein Design | p00000000:"9"
p00000001:"16"
p00000002:"1"
p00000003:"11"
p00000004:"16"
p00000006:"9"
p00000006:"0"
p00000007:"14"
...
p00000047:"13" | title: "871cac30956711eab5bef371aa1bb25a"
user: REDACTED
description:""
objective:"" |
| Vertex AI Text | universal_model_type:
    "single_dense_feature"
token_model_type: "cnn"
token_bow_combiner: "sqrtn"
token_model_type: "bow"
rand:0
batch_size: 4
convnet: "2:3:4*100pa"
dropout_keep_prob: 1
hidden_layer_dims: 50
max_length: 1.54e+03
max_num_classes_for_per_class_metric: 0
max_token_vocab_size: 1e+05
merge_word_embeddings_vocab_size: 1e+05
token_freq_cutoff: 1
tokenizer_spec: "delimiter"
word_embedding: REDACTED
word_embedding_dim: 100 | title: "20220228-621c9aea-0000-2c94"
user: REDACTED
description: REDACTED
objective:"micro-auc-pr-label0_label"
atc_study_dataset_tag:""
atc_study_notes:"" |

