# OpenReview forum: "OmniPred: Language Models as Universal Regressors"
_TMLR — Accepted by TMLR_

### Review · Reviewer_PTPJ · 2024-09-26

**Summary Of Contributions:**

This paper, titled “OmniPred: Towards Language Models as Universal Regressors,” addresses the novel task of using large language models (LLMs) for universal regression tasks across diverse real-world experimental data. The authors propose a new framework, OmniPred, that applies LLMs to perform regression based on textual representations of input-output pairs, demonstrating that LLMs can efficiently handle a wide range of tasks. The paper showcases the model’s strength in transfer learning, the ability to work across multiple domains, and a comparison with traditional regression models, highlighting significant performance improvements.

**Audience:**

Yes

**Broader Impact Concerns:**

I don't think this work involves any ethical implications. The limitation is also well discussed in the paper.

**Claims And Evidence:**

Yes

**Requested Changes:**

It has been mentioned in the weakness part as well:

1. It is suggested to provide a more detailed explanation of OSS Vizier, including its importance in blackbox optimization and how it facilitates the framework in this study. This would make the paper more accessible to a wider audience.

**Strengths And Weaknesses:**

Pro:
1. The task tackled in this paper is significant and impactful. Exploring the potential of language models as universal regressors has broad implications across various fields, including scientific research, engineering, and industrial applications.

2. The use of textual representations for regression is particularly creative and demonstrates the flexibility of LLMs beyond their typical use cases. This method removes the need for complex input feature engineering, which is often required in traditional regression models.

3. The extensive experimental results convincingly show that OmniPred can outperform traditional models like MLPs and boosted trees, especially when trained across multiple tasks. This inspires future exploration of LLMs in similar settings.

4. The multi-task learning approach is well-executed, and the fine-tuning experiments demonstrate the adaptability of the model in unseen tasks, which is crucial in real-world applications.

Cons:
.1 One area where the paper could improve is in its explanation of the OSS Vizier format. The use of this specific open-source software may not be familiar to all readers, particularly those without a background in hyperparameter optimization or blackbox optimization research. More context and explanation would help bridge this gap for the broader audience.

2. While the experiments provide a comprehensive evaluation across tasks, it would be interesting to see how the model performs on high-dimensional regression problems. The complexity of high-dimensional spaces often poses challenges for standard regression models, and understanding OmniPred’s scalability in such cases would strengthen the overall findings.

---

> ### Author Response · Authors · 2024-09-26
> **Follow-up: Thank you!**
>
> Hi Reviewer PTPJ, thanks for the wonderful and positive feedback.
>
> We've updated our draft (changes in blue) to reflect your comments, and as direct responses:
>
> 1. **OSS Vizier and hyperparameter tuning**: Section 4 (Data) now contains an intro paragraph explaining that although many industries contain huge collections of metric data from experiments or systems, these are not typically open-sourced for research, and thus we must use offline hyperparameter tuning trajectories as natural regression datasets. We then added more details about standard search space definitions in blackbox optimization and their implementations in OSS Vizier.
>
> 2. **High dimensional regression problems**: In Figure 6, we have results for "Protein Design" (dimensionality of 125 categories) and "Vertex AI (Tabular)" (dimensionality of 42). For both cases, multi-task OmniPred outperforms the single-task MLP, and even the single-task OmniPred slightly outperforms the MLP.
>
> Thanks again for your time!

---

> > ### Comment · Reviewer_PTPJ · 2024-11-28
> >
> > I appreciated  the authors' response  and have no further questions.

---

### Review · Reviewer_Wdzp · 2024-10-05

**Summary Of Contributions:**

The paper proposes a framework for training language models to perform regression over experimental data, in order to predict metrics, in a multi-task way. The overall aim is to train a "universal regressor" that can predict metrics across various tasks and input spaces using only textual representations, with the application being that ppl can describe tasks in natural language, and would not need to feature engineer or run actual number of trials to get an outcome. The authors test both online and offline training (as in having access to the metric function, online, or just having access to black box input output values from existing datasets of optimization trajectories) and show effectiveness. The also show that the language model-based approach outperforms traditional regression models in many cases, especially when leveraging multi-task learning across different domains, and in terms of out of domain/unseen generalization.
The model can adapt to unseen tasks through fine-tuning while retaining knowledge from previous data.

**Audience:**

Yes

**Claims And Evidence:**

Yes

**Requested Changes:**

Please view the weaknesses. I also think the experimental setup could be polished to make the claims more clear and provide more background on the datasets and tasks.

**Strengths And Weaknesses:**

Strengths:

1. I really like the idea of describing the tasks and taking multiple features and using them as x and y, training in an autoregressive way. It seems novel, and works well. Making regressors easier to use and more accessible could help expand their use case and have non-ML/DS experts build models on their data


Weaknesses:
1. I'm having trouble fully grasping the actual effectiveness/real world applicability of the method. the authors mention reward modeling in RL, yet they do not have any experiments on that. Also, Is this really more computationally efficient than actually training regressors? It seems like the T5 model was trained from scratch, based on section 3. So how universal is this really, if we need to train from scratch? Can't we just dump all of this in the long context of an LLM? do we really need fine-tuning?

2. I also got a bit confused by table 2, and the value y. the meta date m states that y is accuracy, yet it is 1.23. is it %?

---

> ### Author Response · Authors · 2024-10-05
> **Follow-up: Thank you!**
>
> Hi Reviewer Wdzp, we're delighted to hear your positive feedback on our text-to-text regression. We're also very interested in providing data scientists a more user-friendly "production variant" in later work.
>
> We've updated the draft (changes in blue), and to directly address your comments:
>
> 1. These are good questions, with our in-line responses below and also discussions in Section 7.
>   * **Real-World Applicability:** Most industries have very large amounts (terabytes+) of performance-related data, and predicting such metrics would be crucial for their success. These include e.g. airlines and flight demand, real-estate and housing prices, cloud providers and hardware utilization, manufacturing and product quality, and so on. Representing all possible features (especially nested types) as traditional tensors is also very tedious, sometimes impossible. Our work is the first to suggest training text-to-text regressors would be very useful in any of these domains.
>   * **Mentioning RL Reward Modeling:** We meant to say that previous works (ex: RL-HF reward modeling) can be seen as a form of regression but only over ordinal categories, and not high-precision unbounded numbers, whereas our work shows text-to-text numeric regression can be done more generally over natural/experimental objectives. Updated the wording.
>   * **Computational Efficiency:** Since MLPs and GPs are relatively fast, we deliberately made our LM also small for comparison (standard 12-layer encoder-decoder). Performing weight updates over e.g. 1000 single-examples only requires a single GPU and takes at most a minute. There are also multiple ways to make this LM faster for "production" use-cases (quantization, efficient attention, fewer layers). We believe it should be possible for any data scientist to easily use such a model with little compute.
>   * **Needing to train from scratch:** We deliberately initialized from scratch to avoid any confounding issues with English pretraining. It was actually remarkable that the model could still perform regression _without understanding English_. Note also that due to constrained-decoding, regardless of its weights, the model will still always output a number - pretraining is only needed if a user wants to send in offline data from previously similar tasks, which may help for future regression tasks. In Figure 6, we show that single-task regression (without pretraining) is still competitive against traditional baselines.
>   * **Dumping on LLM long-context:** This is a very nuanced discussion which we expand upon in blue (Section 7, In-Context Learning). In short, there are multiple issues with in-context usage:
>     * **Most $(x,m)$ are already too long:** By allowing flexible representations, this means the prompts can have high token-length; for our data, some of our prompts were already reaching up to 8K length. In principle, we might even want to send in an entire configuration file or code associated with the objective, which for our future use-cases, can reach up to 600K token length.
>     * **Finite limit:** Additionally, in-context learning imposes a _finite limit_ on the number of training points at inference. If each prompt is already thousands of tokens long, then the number of in-context points would be very limited, since this is equal to $(\text{max context length}) // (\text{average prompt length})$. Online weight-based tuning (which can be fast, e.g. using LoRA) in principle, allows absorbing unlimited data points.
>     * **Service-based LLMs:** Service-based LLMs such as ChatGPT do not allow fine-tuning over user data pre-inference, which means multi-task training is not possible. This would seriously limit the usage of large amounts of valuable offline evaluation data, which as seen in the paper, can improve inference-time regression significantly.
> 2. **Table 2:** We've edited the wording to say "accuracy (percentage)", and made the example more direct (e.g. stating `<+><7><2><5><E-1>` means $725 \times 10^{-1} = 72.5$.
>
> Experimental setup polished: We've made more clarifying changes in Section 3 throughout, to make it easier to read.
>
> Thank you for your feedback!

---

### Review · Reviewer_pUMR · 2024-11-08

**Summary Of Contributions:**

The paper makes two primary contributions to the field of text-based regression. First, it demonstrates the successful adaptation of transformer architecture and multi-task based pre-training techniques, commonly used in language modeling, to the specific task of mapping textual data to continuous numerical values. Second, through empirical evaluation, it establishes that this approach achieves competitive performance compared to traditional regression methods like Gaussian Processes and Random Forests.

**Audience:**

Yes

**Broader Impact Concerns:**

The broader impact statement of the paper is fine.

**Claims And Evidence:**

Yes

**Requested Changes:**

The empirical evaluation section would benefit from including a comparison with In-Context Learning approaches. Could you add this comparison to strengthen the evaluation?

**Strengths And Weaknesses:**

## Strengths


The paper demonstrates a significant achievement in successfully implementing a pre-training approach without relying on self-supervised data, which is particularly noteworthy given that self-supervision is often considered crucial in large language models.


## Weaknesses

The fine-tuning methodology employed in the study, while functional, remains relatively basic. The paper would benefit from a more comprehensive exploration of advanced post-training optimization techniques. In particular, comparing the current approach with sophisticated methods such as reinforcement learning could provide valuable insights into the model's potential for further improvement.

---

> ### Author Response · Authors · 2024-11-08
> **Follow-Up**
>
> Hi Reviewer pUMR, thanks for taking the time to review and give positive feedback on our paper.
>
> As direct responses to your questions:
>
> 1. **Fine-tuning method:** Our "Online Finetuning" can be considered a form of typical supervised fine-tuning (SFT) over a small dataset we still the standard token cross-entropy (CE) loss.
> 2. **Using RL instead of SFT:** If we understand your intention correctly, it seems the benefit of offline RL (for even pre-training) is to allow non-differentiable rewards (e.g. directly using MSE as the objective, rather than CE as a proxy for MSE). We think this is interesting but worth more investigation in future work.
>     * So far, we've empirically seen that CE loss has a direct monotonic relationship with regression error (i.e. a model with lower validation cross-entropy loss will always have a lower MSE), and thus is a good proxy already.
> 3. **In-Context Learning (ICL):** We discuss these nuances (in blue) in Section 7, but in short, we emphasize that our main message is about multi-task vs single-task training, since maximizing training data is much more important than whether a regressor is ICL or zero-shot. Currently there is no ICL method that is fully multi-task, i.e. simultaneously satisfying: (1) Applicable over very different data formats, and (2) Pretrainable over offline $(x,y)$ data.
>   * **All ICL baselines are single-task:** Thus even if we compared to in-context approaches, the conclusion would remain the same. For example, none of the following ICL baselines are fully multi-task:
>       * Traditional Gaussian Processes (GPs), which do not satisfy either (1) or (2) and are strongly single-task, which we've also shown in our experiments.
>       * LLM servers (Gemini, Claude, etc.), which generally do not allow (2) for users and have limited context window even when allowing (1). (Vacareanu et al, 2024) also showed that at best, they roughly perform similarly to single-task baselines such as random forests.
>       * Transformer-based Neural Processes, which do not satisfy (1) due to fixed tensor dimensions and cannot be applied to our tasks which have different input shapes.
>
> Thanks in advance for the discussion.

---

> > ### Comment · Reviewer_pUMR · 2024-11-10
> > **RL instead of SFT**
> >
> > If I understand correctly, the cross-entropy loss used during both pre-training and SFT phases serves as a differentiable approximation to the true loss defined in equation 1.
> >
> > In cases where y's tokenization consists of a single token, equation 1 can be used directly as the loss function for SFT. However, when y is split into multiple tokens, exact computation of equation 1 becomes computationally challenging, necessitating the use of approximations.
> >
> > While cross-entropy loss is a reasonable choice, one might expect that incorporating both the L1 distance from the ground truth and the sampling operator s could improve the convergence rate.

---

> > > ### Author Response · Authors · 2024-11-10
> > > **Agreed**
> > >
> > > Agreed, and thanks for the suggestion - we've added brief wording on the possibility of RL in Page 4 ("Training") while maintaining page limits.

---

> > ### Comment · Reviewer_pUMR · 2024-11-10
> > **In-Context Learning**
> >
> > Regarding multi-task learning with in-context learning (ICL), there exists a rich body of research examining transformers' ability to learn meta-algorithms that can adapt at test time. For an exemplary study, see Garg et al. (2022), who investigate what function classes transformers can learn in-context.
> >
> > [1] Garg, Shivam, et al. "What can transformers learn in-context? A case study of simple function classes." Advances in Neural Information Processing Systems 35 (2022): 30583-30598.

---

> ### Author Response · Authors · 2024-11-10
> **Previous ICL works are still fixed dimensional**
>
> Apologies if there were any misunderstandings around our phrase "fully multi-task".
>
> We agree that there have been numerous works on the ICL abilities of raw Transformer models, but they have only studied on specifically "_fixed dimensional_ multi-task" regression settings.
>   * For instance, (Garg et al, 2022) in Page 4, "Model Structure" writes that they "map the prompt inputs and outputs into the latent embedding space of the Transformer through a learnable transformation".
>   * Similar settings can be said for other techniques with their specific names, e.g. "Transformer Neural Processes" (Nguyen, 2022).
>
> Issue is these methods are still restrictive with respect to input shapes. Once they've locked into a feature dimension of $d$ (e.g. a regression problem with $d$ continuous variables or a single categorical variable with $d$ choices), it's very difficult to train on a problem any other dimensions using the same set of weights. None of our data can be expressed using a single feature dimension, which makes the "raw-Tensor" class of methods inapplicable for multi-task data in the wild such as ours.
>
> (Chen et al, 2022) tries to resolve this issue by compressing $x$'s using a dedicated vocabulary and moving everything onto the sequence axis, but:
>   * This technique can only be applied to tabular formats (i.e. every input is guaranteed to always have a fixed set of variable values)
>   * Is inapplicable to e.g. combinatorial or conditional inputs where variables might be missing (which occurs in our AutoML data).
>   * Some of our task families also have a very high average variable count (50+, e.g. Protein Design or Vertex AI), which would overload a typical 8192-context window if e.g. the number of data points is above 200.
>
> Although our data from the wild makes a meaningful and clean comparison impossible, we nonetheless agree that it would very interesting to compare against ICL - but this requires a new ICL architecture which is "fully multi-task", specifically resolving all the issues above, which is worth investigating in future work. We hypothesize the right architecture would be an ICL version of OmniPred, i.e.:
>   * Language Encoder to compress strings into representations
>   * Float Decoder to output floats as strings in unnormalized space
>   * ICL Transformer to handle all intermediate tensor representations
>
> Nonetheless, we've also added (Garg et al, 2022) to Page 2 in the "Related Work and Motivation" section.
>
> We hope this clarifies the discussion.

---

### Author Response · Authors · 2024-11-08
**Thanks to all reviewers**

We'd like to again thank all the reviewers for their time and effort in reviewing our paper. We've promptly responded to all reviews early on, and made adjustments to the paper as needed.

During this discussion phase, please let us know if anything else is needed, and we'd be happy to accommodate again. Otherwise, we are delighted from all of the supportive feedback given around this new research topic.

Thanks!

---

### Public Comment · ~Muhammad_Emmad_Siddiqui1 · 2025-01-05
**Can this approach be used for time-series as well?**

Hi, it seems good this technique for regression problems, will it be good for time series problems as well with 30-40 data points, I would love your response on this

---

> ### Author Response · Authors · 2025-01-06
> **Time series naturally is better suited for in-context learning, but...**
>
> Thanks for your interest. At the end of the day, it's really just a question of where your losses will be computed during a regular LLM fine-tuning.
>
> The natural and safe way is to perform in-context learning, and depending on the data format:
>   * Incremental time steps $(m, t_1, t_2, \ldots)$ - you can train autoregressively on $t_1, t_2, \ldots$
>   * Gap in time $(m, t_1, t_3, t_7, t_{42}, \ldots)$ - you will need to a "timestep prompt" $i$ before every $t_i$ and then only compute losses on $t_i$.
>
> But our paper doesn't investigate the in-context case, only basic (prompt, response) scenarios. If following our specific approach literally, then your prompt would be e.g. "What is the value at t=42" and your response is $t_{42}$, and you will need to perform weight-updates at inference time to absorb previous timestep values.
>
> This is much less explored but it would be interesting to see what happens in this "zero-shot" regime.
>
> Hope this helps.

---

### Decision · Action_Editor_uXHG · 2024-12-09

**Recommendation:** Accept as is

**Comment:**

All the reviewers agree that this is an interesting and novel framework for training a flexible regressor using an LLM.  The experiments are compelling, and while not immediately obvious, it is a strength that this method achieves competitive performance with traditional Gaussian Process and Random Forest regressors using a relatively older/lightweight t5 model.

I also appreciate the authors willingness to incorporate the reviewers' feedback, as the paper has made improvements throughout the review process to distinguish what makes OmniPred novel compared to other recent work, e.g., the ICL work of Garg et al. (2022).

Since no reviewers raised the following point, I will accept as is (without minor revision), but I strongly encourage the authors to link to a runnable code example in the camera ready (even if it is just a notebook training the t5 model; note that a more mature package will of course go a longer way).  This will greatly decrease the barrier for readers to use this work in their own research and can significantly increase the impact of the paper.

**Audience:**

Yes

**Claims And Evidence:**

Yes

---

> ### Author Response · Authors · 2024-12-23
> **Submitted Camera Ready Version with Code**
>
> Hi AE, thanks for the acceptance decision!
>
> We've submitted the camera ready version, along with a Github link to the official codebase used. We also added a video presentation (from an earlier talk) for the paper.